# Safety, Adequacy, and Accuracy of Transvaginal Ultrasound-Guided Tru-Cut Biopsy of Gynaecologic Tumours: A Retrospective Single-Centre Study

**DOI:** 10.3390/diagnostics15091133

**Published:** 2025-04-29

**Authors:** Munachiso Iheme Ndukwe, Dominik Karasek, Denisa Pohanková, Ivan Praznovec, Petra Bretova, Martin Stepan, Dominik Habes, Jan Laco, Akaninyene Eseme Ubom, Milan Vosmik, Igor Sirak

**Affiliations:** 1Department of Obstetrics and Gynaecology, University Teaching Hospital Hradec Kralove, Charles University, Faculty of Medicine in Hradec Kralove, 500 05 Hradec Kralove, Czech Republic; munachiso.ndukwe@fnhk.cz (M.I.N.); dominik.karasek@fnhk.cz (D.K.); ivan.praznovec@fnhk.cz (I.P.); bretova.petra@gmail.com (P.B.); mstepan.hk@post.cz (M.S.); dominik.habes2@fnhk.cz (D.H.); 2Department of Oncology and Radiotherapy, University Teaching Hospital Hradec Kralove, Charles University, Faculty of Medicine in Hradec Kralove, 500 05 Hradec Kralove, Czech Republic; denisa.pohankova@fnhk.cz (D.P.); milan.vosmik@fnhk.cz (M.V.); igor.sirak@fnhk.cz (I.S.); 3The Fingerland Department of Pathology, University Teaching Hospital Hradec Kralove, Charles University, Faculty of Medicine in Hradec Kralove, 500 05 Hradec Kralove, Czech Republic; lacoj@lfhk.cuni.cz; 4Department of Obstetrics, Gynaecology and Perinatology, Obafemi Awolowo University Teaching Hospitals Complex, Ile-Ife 220882, Nigeria

**Keywords:** cancer, Czech, gynaecologic cancers, malignancy, ultrasound, women cancers

## Abstract

**Objective:** To evaluate the safety, adequacy, and accuracy of tru-cut biopsy of gynaecologic tumours in a population of Czech women. **Methods:** A four-year retrospective study of ultrasound-guided tru-cut biopsy of gynaecologic tumours was conducted in the Department of Obstetrics and Gynaecology, Charles University, Hradec Kralove, Czech Republic. **Results:** One hundred and four women with gynaecologic tumours underwent transvaginal tru-cut biopsy within the study period. The most common indication for tru-cut biopsy in more than one-half of the women was a suspicion of malignancy/inability to exclude malignancy (59, 56.7%). Most of the tumours were malignant on histopathological examination (71, 68.3%), with advanced ovarian cancer being the most common type of malignancy (43/71, 60.6%). The overall adequacy and accuracy rates of tru-cut biopsy were 93.3% and 93.3%, respectively. Most of the inadequate samples were obtained from overweight and obese women (5/7, 71.4%), with only one biopsy sample taken in the majority of the inadequate biopsies (5/7, 71.4%). Accuracy was higher for malignant than benign tumours (97.7% vs. 82.4%). For malignant tumours, accuracy was highest for advanced ovarian cancers (33/40, 82.5%). Only one case was complicated by bleeding, giving an overall complication rate of 1%. The complicated biopsy was taken by a gynae-oncology trainee. **Conclusions:** Tru-cut biopsy is a cost-effective and safe preoperative diagnostic modality for patients with gynaecologic tumours, offering high adequacy and accuracy. It is particularly useful in patients with advanced ovarian cancer, most of whom present late with inoperable tumours that contraindicate primary surgery.

## 1. Introduction

Tru-cut biopsy is a minimally invasive technique to obtain cytological or tissue specimens for the histopathological diagnosis of malignant lesions [1]. It is indicated to make preoperative diagnoses to plan further treatment when primary surgery is not indicated such as in patients who are unfit for surgery due to comorbidities, advanced and inoperable disease, in patients with lesions suspected to be malignant or recurrent, and patients with tumours of uncertain origin [2,3].

Tru-cut biopsy has significant advantages over other minimally invasive biopsy techniques. In contrast to fine-needle aspiration cytology (FNAC), which allows only cytological evaluation of predominantly cystic lesions, and fine-needle aspiration biopsy (FNAB), which collects smaller tissue samples that are not always architecturally preserved with only occasional three-dimensional tissue fragments, tru-cut biopsy allows for a larger tissue to be obtained, ensuring the involvement of architecturally preserved tissue samples and allowing for other histological examinations such as immunohistochemistry [2,3,4]. It is a simple, safe, and cost-effective biopsy method that can be performed in an outpatient setting without the need for any special patient preparation or general anaesthesia, and also eliminates the risks of morbidity and mortality from surgery as well as any delays in further treatment [5,6].

The use of radiological modalities such as ultrasound, computed tomographic (CT) scan, and magnetic resonance imaging (MRI) for the direct visualization of the biopsy needle during the procedure minimizes the risks of complications. Colour Doppler allows for the selection of the most appropriate tissue area for biopsy, avoiding puncturing large blood vessels or tissue areas with high vascularity [2,6]. Compared to CT scan and MRI, ultrasound is more widely available, does not radiate, and offers more flexibility in terms of the route of biopsy, such as transabdominal, transvaginal, or transrectal [2]. The transvaginal ultrasound route allows biopsy of pelvic masses that may be difficult to access transabdominally without the high risks of vascular and intestinal injuries that are associated with the transabdominal route, and also biopsy of lymph nodes that are difficult to access by other routes [6]. The reported diagnostic accuracy of tru-cut biopsy is 76–99% [1,5,7]. While tru-cut biopsy for breast and prostate pathologies is widely reported in the literature, it is much less so for gynaecological lesions [5,7,8]. This study sought to evaluate the safety, adequacy, and accuracy of transvaginal tru-cut biopsy in gynaecologic tumours in a population of Czech women.

## 2. Materials and Methods

### 2.1. Study Design and Setting

This retrospective study of ultrasound-guided transvaginal tru-cut biopsy of gynaecologic tumours was conducted in the Department of Obstetrics and Gynaecology, Charles University, Hradec Kralove, as part of a post-graduate project.

### 2.2. Inclusion and Exclusion Criteria

All patients with gynaecologic tumours who underwent ultrasound-guided transvaginal tru-cut biopsy in the department from January 2018 to December 2021 were included in this study.

### 2.3. Data Collection

The baseline characteristics of the patients, including age, menopausal status, body mass index (BMI), and previous history of and type of previous cancer, were obtained from the electronic medical records of the patients using a purpose-designed proforma. Clinical parameters including CA-125 and HE-4 levels, risk of ovarian malignancy algorithm (ROMA) index, indication for tru-cut biopsy, radiological characteristics of the tumour (including colour score), route and site of biopsy, complications of biopsy, and histopathological diagnoses of tru-cut biopsy samples and postoperative samples, were also obtained and recorded. The ROMA index is a predictive algorithm that incorporates CA-125, HE-4, and menopausal status to distinguish between benign and malignant ovarian masses [9]. Colour score is used to describe the amount of blood flow to a tumour. Colour score 1 means no detectable blood flow; score 2 means minimal blood flow detected; score 3 means moderate blood flow; and score 4 means highly vascular/very strong flow [10].

### 2.4. Procedure for Tru-Cut Biopsy

Ultrasound-guided tru-cut biopsy in the department was introduced by Peter Skapinec, an experienced gynaecological oncologist, who, along with I.P., a trainee in gynaecological oncology, performed all biopsies included in this study. The procedure was initiated with a combination of transvaginal and transabdominal ultrasound, using a GE Voluson E8 machine (BT13) (GE Healthcare Austria GmbH, Zipf, Austria), with the patient positioned in the lithotomy position during the procedure. A Palium^®^ biopsy gun (BIP Evocore EC2215, BIP GmbH, Türkenfeld, Germany) with a disposable 14–20 G tru-cut needle was employed, and the needle was connected to the ultrasound probe via a needle guide for the transvaginal approach. Doppler ultrasound was employed to guide the tru-cut needle during the procedure. Typically, one to two biopsy samples were collected, although a third and fourth sample was obtained in a small number of cases, depending on the operator′s clinical judgement. No anaesthesia was used during the transvaginal approach.

### 2.5. Analysis of Adequacy, Accuracy, and Safety

Adequacy: Tissue samples that allowed histological evaluation including immunohistochemistry were considered adequate [2,7].

Accuracy: This was assessed as the agreement/concordance of tru-cut biopsy histopathological diagnoses and postoperative histopathological diagnoses in patients that underwent surgery [2,7]. Patients that had inadequate tissue samples on tru-cut biopsy were excluded from evaluation for accuracy.

Safety: Safety was assessed based on complication rate [2,7].

### 2.6. Statistical Analysis 

Data obtained were analysed using IBM Statistical Product and Service Solutions (SPSS) Statistics for Windows, version 25 (IBM Corp., Armonk, NY, USA). Missing data were excluded from analysis. Frequencies and percentages are presented in tables. Normally distributed data are presented as mean ± SD, while skewed/non-normally distributed data are expressed as the median (interquartile range; IQR).

## 3. Results

One hundred and four women with gynaecologic tumours underwent transvaginal tru-cut biopsy within the study period of 2018–2021.

### 3.1. Baseline Characteristics of the Women Who Underwent Transvaginal Tru-Cut Biopsy

The mean age of the women was 61.6 ± 12.1 years, with an age range of 26–84 years. Their mean body mass index (BMI) was 27.0 ± 6.4 kg/m^2^ (overweight range), with a median CA-125, HE4, and ROMA score of 180.0 (54.4–836.9) mIU/mL, 151.4 (57.9–527.9) pmol/L, and 73.2 (15.3–95.1), respectively (Table 1). Most of the women were postmenopausal (86, 82.7%), with no previous personal history of cancer (72, 69.2%). Of those who had a previous history of malignancy, the majority (19/32, 59.4%) were of genital tract origin (Table 1). More than a third (5/13, 38.5%) of the non-genital malignancies in women who had a previous history of malignancy were breast cancer.

### 3.2. Indications for Transvaginal Tru-Cut Biopsy and Radiological Characteristics of Tumours

The most common indication for transvaginal tru-cut biopsy in more than one-half of the women was a suspicion of malignancy/inability to exclude malignancy (59, 56.7%). An expert ultrasound scan was the primary imaging modality in the majority of cases (75, 72.1%). The median largest tumour diameter was 71.7 (44.0–110.0) mm, with most of the tumours being solid and multilocular (66/95, 69.5%), with irregular margins (55/89, 61.8%) and colour scores of 3–4 (27/35, 77.1%) (Table 2). Ascites was present in less than one-half of cases (45/103, 43.7%).

### 3.3. Transvaginal Tru-Cut Biopsy Procedure Characteristics and Histological Diagnoses

A pelvic mass was biopsied in three-fourths of cases (79, 76.0%). Sizes 16–18 G needles were used in the majority of cases (81/88, 92.0%), with size 14 G used in 6/88 (6.8%) and size 20 G used in only 1 case. A single or two samples were taken in most cases (95, 91.3%) (Table 3). Slightly more than three-fourths (80, 76.9%) of the biopsies were performed by the experienced gynae-oncologist, while less than one-fourth (24, 23.1%) were performed by the gynae-oncology trainee.

Most of the tumours were malignant on histopathological examination (71, 68.3%), with advanced ovarian cancer being the most common type of malignancy (43/71, 60.6%) (Table 3). Recurrent tumours constituted one-tenth of cases (11, 10.6%). Most of the malignant ovarian cancers were serous cystadenocarcinomas (36/43, 83.7%), while others included clear cell carcinomas (5/43, 11.6%), seromucinous carcinoma (1/43, 2.3%), and mucinous cystadenocarcinoma (1/43, 2.3%).

### 3.4. Adequacy and Safety of Transvaginal Tru-Cut Biopsy

The overall adequacy rate of transvaginal tru-cut biopsy was 93.3%. A non-diagnostic/inadequate sample was seen in only 6.7% (7) of cases, with pelvic mass biopsies constituting nearly three-fourths (5/7, 71.4%) of the inadequate/non-diagnostic samples. Slightly more than 70% (5/7) of the inadequate samples were obtained from overweight and obese women, whereas of the women that had adequate samples, only 54.6% (53/97) were overweight or obese. All of the inadequate biopsies were performed by the experienced gynae-oncologist. Most (5/7, 71.4%) of the procedures that were inadequate had only one biopsy sample taken.

Only one case was complicated by bleeding, giving an overall complication rate of 1.0%. The complication occurred in an overweight woman with a BMI of 28.7 kg/m^2^, who had a transvaginal cervical biopsy of a metastatic non-genital tract malignant tumour. The procedure was performed by the gynae-oncology trainee using a size 16 G needle.

### 3.5. Definitive Postoperative Histological Diagnoses and Accuracy of Transvaginal Tru-Cut Biopsy

Sixty-four (61.5%) women had surgery followed by postoperative definitive histological diagnoses. Of these, 15 (23.4%) were benign tumours, while 49 (76.6%) were malignant. The predominant malignant diagnoses were ovarian cancers (40/49, 81.6%), while endometrial cancers (3/49, 6.1%) were the least common (Table 4). Two of the malignant cancers were recurrent tumours, one was ovarian cancer and the other metastasis from a non-genital tract cancer.

Overall accuracy of transvaginal tru-cut biopsy was 93.3%, with the final postoperative histological diagnoses in 56 of the 64 operated women being in agreement with the tru-cut biopsy histological diagnoses. In total, 4 of the 64 operated women had inadequate/non-diagnostic tru-cut biopsies, and were excluded from evaluation of accuracy. For benign tumours, the concordance rate of definitive postoperative histological diagnoses with tru-cut biopsy histological diagnoses was 82.4% (14/17), while the concordance rate of definitive postoperative malignant histological diagnoses with tru-cut biopsy malignant histological diagnoses was 97.7% (42/43). By cancer type, the concordance rate of postoperative/definitive histological diagnoses with tru-cut biopsy histological diagnoses was the highest for advanced ovarian cancers (33/40, 82.5%), followed by metastases from non-genital tract cancers (4/6, 66.7%), and only one-third (1/3, 33.3%) for advanced endometrial cancers. Recurrent cancers had 100% (2/2) concordance of postoperative/definitive diagnoses and tru-cut biopsy histological diagnoses.

In total, 39 (69.6%) of the 56 accurate tru-cut biopsies were taken by an experienced gynae-oncologist, while 17 (30.4%) were taken by a gynae-oncology trainee. Of the 60 operated women that had adequate tru-cut biopsies, 41 of the tru-cut biopsies were performed by an experienced gynae-oncologist, and 19 by a gynae-oncology trainee. In total, 39 of the 41 (95.1%) adequate tru-cut biopsies performed by an experienced gynae-oncologist were accurate, and 17 of the 19 (89.5%) adequate tru-cut biopsies performed by a gynae-oncology trainee were accurate. The difference was not statistically significant (*p* = 0.415).

## 4. Discussion

The tru-cut biopsy accuracy of 93.3% found in this study is higher than the 88.2% and 90.2% reported by Buonomo et al. (2022) [2] and Vlasak et al. (2020) [11], respectively. Our accuracy is, however, lower than the accuracy of 97.5% documented by Asp et al. (2023) [1] and 98% by Kar et al. (2018) [12]. The largest study comparing fine-needle aspiration biopsy and tru-cut biopsy by Chojniak et al. (2006), which included 1300 CT-guided biopsies from the chest, abdomen, retroperitoneum, and head/neck regions, found a diagnostic accuracy rate of 82–100% for tru-cut biopsy [13]. Our study accuracy rate falls within this range. A diagnostic inaccuracy of 12.8% between tru-cut biopsy histology and the final histological examination was reported by Lengyel et al. (2021) [14]. The tru-cut biopsy adequacy of 93.3% in this study is within the ranges of 91–95% reported by Kar et al. (2018) [11] and 93–100% documented by Chojniak et al. (2006) [13], but higher than the adequacy of 80.2% in the study by Verschuere et al. (2021) [5].

The adequacy and accuracy of tru-cut biopsy are affected by the site and origin of tumours, tumour heterogeneity, sampling errors, and differential diagnostic challenges [2,14]. Sampling errors can be reduced by avoiding taking biopsies from cystic or necrotic portions of tumours. Colour Doppler imaging can help guide the biopsy needle to viable portions of the tumour [8]. The median tumour diameter of 7.2 cm in this study was a favourable factor for the high biopsy adequacy recorded. Lin et al. (2017) reported that adequacy is significantly increased when a tumour is >2 cm in diameter [15]. Adequacy is also affected by the number of tru-cut biopsy samples taken. Verschuere et al. (2021) found that when a single tru-cut biopsy tissue cylinder was taken, the adequacy was 75%, increasing to 94.4% with two cylinders, and 100% for three or four cylinders [5]. In our study, 1–2 samples were taken in more than 90% of the tru-cut biopsies, with 3–4 samples taken in <10% of cases. More than 70% of the tru-cut biopsies that were inadequate in this study had only one sample taken.

Slightly more than 70% of women with inadequate tru-cut biopsies in this study were overweight/obese. Although this could be explained by the fact that more than 70% of the tru-cut biopsies in this study were ultrasound-guided, and obesity impedes the accuracy of ultrasound [16], in contrast to our study finding, Zikan et al. (2010) [7] and Vershuere et al. (2021) [5] did not find any association between obesity and the adequacy of tru-cut biopsy. The average BMI of our study population was in the overweight range, and more than 70% of the inadequate samples were biopsies of pelvic masses, as similarly reported by Asp et al. (2023) [1]. For obese patients and for sampling of pelvic tumours, the transvaginal route is recommended and safer than the transabdominal route, which carries a high risk of inadvertent intestinal injury [17]. Access to and biopsy adequacy for pelvic tumours is higher with a transvaginal approach, as these tumours are close to the vagina, allowing for adequate sampling even in extreme obesity [5,7]. We utilised the transvaginal route for all the tru-cut biopsies in this study. Curiously, all of the inadequate tru-cut biopsies in this study were obtained by an experienced gynae-oncologist. This was probably owing to the fact that the trainee performed the less difficult tru-cut biopsies, performing less than 25% of all the tru-cut biopsies in this study.

Our study demonstrated that tru-cut biopsy is very safe, with a complication rate of 1%. Gao et al. (2019) recorded no complication in their study [18]. Fischerova et al. (2008) [8] reported no complication with transvaginal tru-cut biopsy and a single complication in the transabdominal group. These findings have been corroborated by other authors [19,20,21]. Bleeding during a tru-cut biopsy can be caused by injury to the tumour itself or to an intra-abdominal organ. This risk can be minimized by the use of Doppler imaging. Patient characteristics such as thrombocytopenia, coagulopathies, or anticoagulant therapy can increase the risk of bleeding, especially from highly vascular tumours, and so the presence of these conditions should contraindicate tru-cut biopsy [8].

In this study, 61% percent of the malignant tru-cut biopsies were indicated for advanced ovarian cancer, with 20% of the patients considered either unfit for surgery or with inoperable tumours. Ovarian cancer is the fourth most common cause of cancer-related mortality in women and the most lethal of all gynaecological malignancies [22]. Most cases are not diagnosed until the disease is advanced, with about 30% being inoperable at the time of diagnosis [11]. Primary surgery is not indicated in such cases, but rather neoadjuvant chemotherapy to reduce the tumour bulk and increase operability. Preoperative tissue histological diagnosis is required to plan such treatment, and tru-cut biopsy plays a significant diagnostic role in such situations. Anwer et al. (2005) [23] compared laparoscopic and image-guided tru-cut biopsy in the diagnosis of ovarian cancer and reported that adequate biopsy was obtained without any complications with tru-cut biopsy compared to three complications of port-site hematoma, uterine perforation, and anaesthetic complication in the laparoscopic group. The cost ratio was 1:7.2 in favour of imaging-guided tru-cut biopsy. Furthermore, laparoscopy requires general anaesthesia, with associated risks.

## 5. Conclusions

Tru-cut biopsy is a safe preoperative diagnostic modality for patients with gynaecologic tumours, offering high adequacy and accuracy. It is cost-effective and can be performed in an outpatient setting without the need for general anaesthesia. Tru-cut biopsy is particularly useful in patients with advanced ovarian cancer, most of whom present late with inoperable tumours that contraindicate primary surgery. In such patients, the biopsy allows for a preoperative tissue histological diagnosis, helping to plan neoadjuvant chemotherapy to reduce tumour bulk and improve operability.

## Figures and Tables

**Table 1 diagnostics-15-01133-t001:** Baseline characteristics of women who underwent transvaginal tru-cut biopsy.

Characteristic	Frequency, *n* = 104	Percentage (%)
**Age (years)**		
<40	4	3.8
40–49	17	16.4
50–59	23	22.1
≥60	60	57.7
**Menopausal status**		
Premenopausal	18	17.3
Postmenopausal	86	82.7
**Mean BMI (kg/m^2^)**	27.0 ± 6.4	
**Median CA-125 (mIU/mL)**	180.0 (54.4–836.9)	
**Median HE (pmol/L)**	151.4 (57.9–527.9)	
**Median ROMA score**	73.2 (15.3–95.1)	
**Previous history of cancer**		
Yes	32	30.8
No	72	69.2
**Type of previous cancer (*n* = 32)**		
Genital malignancy	19	59.4
Non-genital malignancy	13	40.6

**Table 2 diagnostics-15-01133-t002:** Indications for transvaginal tru-cut biopsy and radiological characteristics of tumours.

Characteristics	Frequency	Percentage (%)
**Indication for tru-cut biopsy (*n* = 104)**		
Malignancy could not be excluded	59	56.7
Suspicion of recurrence	19	18.3
Not suitable for surgery	11	10.6
Suspicion of genital tract inoperable tumour	10	9.6
Suspicion of primary non-genital tract tumour	5	4.8
**Imaging modality (*n* = 104)**		
Ultrasound	75	72.1
CT scan	27	26.0
MRI	2	1.9
**Radiological characteristics of tumour**		
*Median largest tumour diameter (mm)*	*71.7 (44.0–110.0)*	
*Type of tumour (n = 95) **		
Unilocular solid	29	30.5
Multilocular solid	66	69.5
*Tumour margin (n = 89) **		
Regular	34	38.2
Irregular	55	61.8
*Colour score (n = 35) **		
1	7	20.0
2	1	2.8
3	10	28.6
4	17	48.6
*Presence of ascites (n = 103) **		
Yes	45	43.7
No	58	56.3

* *n* ≠ 104 owing to missing records/non-documentation.

**Table 3 diagnostics-15-01133-t003:** Transvaginal tru-cut biopsy procedure characteristics and histological diagnoses.

Characteristic	Frequency, *n* = 104	Percentage (%)
**Site of biopsy**		
Pelvic mass	79	76.0
Vaginal vault/cuff	14	13.5
Cervix	9	8.6
Omental cake	2	1.9
**Number of samples taken**		
1	63	60.6
2	32	30.8
3	7	6.7
4	2	1.9
**Histological diagnoses**		
Benign	26	25.0
Malignant	71	68.3
Non-diagnostic sample	7	6.7
**Type of malignant tumour (*n* = 71)**		
Advanced ovarian cancer	43	60.6
Metastasis from non-genital tract cancer	10	14.1
Advanced endometrial cancer	8	11.3
Cervical cancer	5	7.0
Uncertain aetiology	5	7.0

**Table 4 diagnostics-15-01133-t004:** Postoperative definitive histological diagnoses.

Characteristic	Frequency	Percentage (%)
**Definitive Postoperative Histology (*n* = 104)**		
Yes	64	61.5
No	40	38.5
**Definitive Postoperative Histological Diagnoses (*n* = 64)**		
Benign	15	23.4
Malignant	49	76.6
**Type of malignancy (*n* = 49)**		
Advanced ovarian cancer	40	81.6
Metastases from non-genital cancer	6	12.3
Advanced endometrial cancer	3	6.1

## Data Availability

The raw data supporting the conclusions of this article will be made available by the authors on request.

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
