# Peer review of "Safety, Adequacy, and Accuracy of Transvaginal Ultrasound-Guided Tru-Cut Biopsy of Gynaecologic Tumours: A Retrospective Single-Centre Study"

_diagnostics, 2025, doi:10.3390/diagnostics15091133_

Round 1

Reviewer 1 Report

Comments and Suggestions for Authors

Dear authors,

Thanks for your work

A few comments to be noted

1- The accuracy of trucut biopsy over FNAC/B is well established through many published manuscripts.  In its current form, the manuscript -with all respect to the authors' efforts- does not add to the literature

2- Confining the results to the transvaginal route could have been better and the idea could be "Safety, Adequacy, and Accuracy of Transvaginal Ultrasound-Guided Tru-Cut Biopsy of Gynaecologic Tumours: A Retrospective Single-Center Study."

3- Why the term "tru-cut" was used instead of "core needle biopsy"?4

4- Please analyze all the results regarding the effect of the radiologist's experience on the biopsy accuracy

5- Why was the period from 2018-2021 only chosen?

6- The rate of using MRI is too small. This should have been explained as MRI is the best method to help distinguish benign from malignant ovarian lesions

7- The pathological variants of ovarian cancer are not mentioned in the results. Please add them

8- Please discuss in detail how your work does not contradict the guidelines for the diagnosis of ovarian cancer which prohibit trucut biopsy in early tumors.

9- In patients with endometrial cancer, why was D&C not performed?

Author Response

Comment 1: The accuracy of trucut biopsy over FNAC/B is well established through many published manuscripts.  In its current form, the manuscript -with all respect to the authors' efforts- does not add to the literature.

Response: Thank you.

Comment 2: Confining the results to the transvaginal route could have been better and the idea could be "Safety, Adequacy, and Accuracy of Transvaginal Ultrasound-Guided Tru-Cut Biopsy of Gynaecologic Tumours: A Retrospective Single-Center Study."

Response: The data has now been confined to transvaginal ultrasound-guided tru-cut biopsy of gynecologic tumours and re-analysed as such. The title has also been changed as recommended. All changes are highlighted in red fonts throughout the paper. Thank you.

Comment 3: Why the term "tru-cut" was used instead of "core needle biopsy"?

Response: Truc-cut and core needle biopsy are synonymous and can be used interchangeably. In our setting and hospital, we use tru-cut. Thank you.

Comment 4: Please analyze all the results regarding the effect of the radiologist's experience on the biopsy accuracy

Response: This analysis has been done and presented in the last paragraph of the Results section just before the Discussion on Page 15. Thank you.

Comment 5: Why was the period from 2018-2021 only chosen?

Response: This was the period during which the most tru-cut biopsies were performed. Thank you.

Comment 6: The rate of using MRI is too small. This should have been explained as MRI is the best method to help distinguish benign from malignant ovarian lesions

Response: This is very well noted. Thank you.

Comment 7: The pathological variants of ovarian cancer are not mentioned in the results. Please add them

Response: The pathological variants of ovarian cancer have now been mentioned in the Results section under "Transvaginal tru-cut biopsy procedure characteristics and histological diagnoses" (Page 11). Thank you.

Comment 8: Please discuss in detail how your work does not contradict the guidelines for the diagnosis of ovarian cancer which prohibit trucut biopsy in early tumors.

Response: All the tru-cut biopsies for ovarian cancer in this study were done for advanced ovarian cancer and not early tumours. Thank you. 

Comment 9: In patients with endometrial cancer, why was D&C not performed?

Response: Being a retrospective study, we could not find the reason for this from the case records. We recognise this as a limitation of a retrospective study design. Thank you. 

Reviewer 2 Report

Comments and Suggestions for Authors

The title "Safety, Adequacy, and Accuracy of Ultrasound-Guided Tru-Cut Biopsy of Gynaecologic Tumours: A Retrospective Single-Center Study" is adequate for the presented outcomes, the article is well-written, and it shows the outcomes of true-cut biopsy of gynaecologic tumours in Czech population. However, the introduction needs complementation. 

Below there are some suggested corrections:

Line 22: "within the study period"— add the years of 2018-2021, please.

Line 32: "The complicated biopsy was taken by a gynae-oncology trainee"—I think it is unnecessary here.

Line 50: "... and fine needle aspiration biopsy (FNAB) that collects smaller tissue samples that are not always architecturally preserved..." - FNAB sample is usually used to obtain cytology, and the 3-dimensional tissue fragments are found occasionally.

Introduction: Describe ROMA and colour score, please.

Line 171: "procedure was performed by the gynae-oncology trainee" and size of a needle and information on the cause of bleeding (needle forged to the vessels/coagulation disorders).

Results: Add an exemplary ultrasound images from benign and malignant biopsies, please.

Comments on the Quality of English Language

Line 33: "was highest" -> was the highest

Line 45: "comorbidities or advanced or inoperable"—in English language "or" means only one option is true; please change "or" to a comma or "and".

Line 95: "14-20G" -> 14-20 G

Discussion: Correct references, please (e.g., "reported by Kar et al [9]" -> reported by Kar et al. (2020) [10]").

Author Response

Comment 1: The title "Safety, Adequacy, and Accuracy of Ultrasound-Guided Tru-Cut Biopsy of Gynaecologic Tumours: A Retrospective Single-Center Study" is adequate for the presented outcomes, the article is well-written, and it shows the outcomes of true-cut biopsy of gynaecologic tumours in Czech population. However, the introduction needs complementation. 

Response: Thank you.

Comment 2: Line 22: "within the study period"— add the years of 2018-2021, please

Response: This has now been added (Page 7). Thank you.

Comment 3: Line 32: "The complicated biopsy was taken by a gynae-oncology trainee"—I think it is unnecessary here.

Response: This has been removed. Thank you.

Comment 4: Line 50: "... and fine needle aspiration biopsy (FNAB) that collects smaller tissue samples that are not always architecturally preserved..." - FNAB sample is usually used to obtain cytology, and the 3-dimensional tissue fragments are found occasionally.

Response: This has been modified in the Introduction on Page 4. Thank you.

Comment 5: Introduction: Describe ROMA and colour score, please.

Response: These have now been described with references in the materials and methods section under "Data collection" (Pages 5-6). Thank you.

Comment 6: Line 171: "procedure was performed by the gynae-oncology trainee" and size of a needle and information on the cause of bleeding (needle forged to the vessels/coagulation disorders).

Response: We have now included information on the size of needle (Page 13) but could not find the cause of bleeding from the case records. Thank you.

Comment 7: Results: Add an exemplary ultrasound images from benign and malignant biopsies, please

Response: As much as we could have liked to add these, we could not find any for our study population being a retrospective study. We recognise this is a limitation of a retrospective study design. Thank you. 

Comment 8: Line 33: "was highest" -> was the highest

Response: This has been corrected. Thank you.

Comment 9: Line 45: "comorbidities or advanced or inoperable"—in English language "or" means only one option is true; please change "or" to a comma or "and".

Response: This has been corrected. Thank you.

Comment 10: Line 95: "14-20G" -> 14-20 G

Response: This has been corrected. Thank you

Comment 11: Discussion: Correct references, please (e.g., "reported by Kar et al [9]" -> reported by Kar et al. (2020) [10]").

Response: The references have been corrected. Thank you.

Round 2

Reviewer 1 Report

Comments and Suggestions for Authors

Dear authors,

Thanks very much for your replies and the modifications performed accordingly

Still, there are two points that need to be explained

1- The authors explained choosing the period from 2018-2021 only because this was the period during which the most tru-cut biopsies were performed. Do you mean that you stopped using this method later on? If so, why was stopped despite being adequate and safe and if not, why the recent results are not included in the study

2- The explanation for the very small rate of using MRI is still not available

Author Response

Comment 1: The authors explained choosing the period from 2018-2021 only because this was the period during which the most tru-cut biopsies were performed. Do you mean that you stopped using this method later on? If so, why was stopped despite being adequate and safe and if not, why the recent results are not included in the study

Authors' response: Tru-cut biopsies were introduced and started in our unit in 2018.  The gynae-oncologist, who had the expertise and who introduced the procedure in our unit, subsequently left our hospital in 2021, and we since stopped performing tru-cut biopsies. Our department is currently about restarting the procedure and thought to perform an audit of all the procedures performed between 2018-2021 to guide future performance and practice. Thank you.

Comment 2: The explanation for the very small rate of using MRI is still not available

Authors' response: The primary investigative and staging modalities for ovarian tumours, which were the predominant tumours in our study, are ultrasound scan and CT scan. This explains the small rate of MRI in our study. Thank you

Reviewer 2 Report

Comments and Suggestions for Authors The manuscript has been finely corrected. It is valuable to know the performance of medical procedures worldwide to improve healthcare.

Author Response

Comment: The manuscript has been finely corrected. It is valuable to know the performance of medical procedures worldwide to improve healthcare

Authors' response: Thank you for your very valuable inputs and feedback, which has significantly improved our manuscript.